# Immunohistochemical Expression of Neurokinin-A and Interleukin-8 in the Bronchial Epithelium of Horses with Severe Equine Asthma Syndrome during Asymptomatic, Exacerbation, and Remission Phase

**DOI:** 10.3390/ani11051376

**Published:** 2021-05-12

**Authors:** Maria Morini, Angelo Peli, Riccardo Rinnovati, Giuseppe Magazzù, Noemi Romagnoli, Alessandro Spadari, Marco Pietra

**Affiliations:** 1Department of Veterinary Medical Sciences, University of Bologna, 40064 Bologna, Italy; angelo.peli@unibo.it (A.P.); riccardo.rinnovati2@unibo.it (R.R.); noemi.romagnoli@unibo.it (N.R.); alessandro.spadari@unibo.it (A.S.); marco.pietra@unibo.it (M.P.); 2DVM, Vet Practitioner, 40024 Castel San Pietro Terme, 40064 Bologna, Italy; magazzu.consultant@gmail.com

**Keywords:** respiratory tract, horse, severe equine asthma, immunohistochemistry, IL-8, NKA

## Abstract

**Simple Summary:**

Severe equine asthma (EA) syndrome, formerly termed Recurrent Airway Obstruction (RAO) or heaves, is one of the most common respiratory diseases in adult horses and a frequent cause of poor equine performance. The affected animals may show periods of clinical remission followed by periods of exacerbation over months to years. Therefore, the aim of the present study was to investigate the histological features, and the Neurokinin-A (NKA) and Interleukin-8 (IL-8) immunoreactivity on bronchoscopic biopsies in horses, obtained during different phases of the disease (asymptomatic, exacerbation and remission). Histological samples of EA-affected horses appeared significantly different from those of non-EA-affected horses (control group) throughout the experimental phase, from inclusion to exacerbation and remission, and intensity of NKA immunopositivity of horses with severe EA was significantly higher than that of control horses in late exacerbation and in remission phase. No significant difference between horses with severe EA in each phase and control horses was noticed for IL-8 immunoreactivity. Moreover, no influence of bronchial sampling position on histological and immunohistochemistry results was found, and it suggests that bronchial structural and functional modification during severe equine asthma tends to be distributed homogeneously throughout the respiratory tree.

**Abstract:**

Severe equine asthma (EA) syndrome is a chronic obstructive disease characterized by exaggerated contraction, inflammation, and structural alteration of the airways in adult horses, when exposed to airborne molds and particulate material. However, little is known about the relationship between the degree and type of inflammation on one hand, and the severity of the disease and the response to treatment on the other. Furthermore, to date, very few studies evaluate the diagnostic value of histology and immunohistochemical features of endoscopic biopsies on subjects with severe equine asthma. To investigate the expression of two inflammatory markers (NKA and IL-8) before, during, and after the exacerbation of severe EA, a histological and immunohistochemical study was carried out on a series of biopsy samples collected by bronchoscopy from six EA-affected horses subjected to process exacerbation through environmental stimuli and then to pharmacological treatment. The application of a histological biopsy scoring system revealed a significant difference between control cases and the EA-affected horses in all experimental phases (asymptomatic, early exacerbation phase, late exacerbation phase, and remission phase). For immunohistochemistry (IHC), only the intensity of NKA positivity increases significantly between control horses and the EA horses at late exacerbation and remission phases. In EA-affected horses, a difference was detected by comparing histology between asymptomatic and remission phase, meanwhile, NKA and IL-8 showed no differences between the experimental phases. Based on these results we can assert that: (1) The endoscopic biopsies generate reliable and homogeneous samples in the entire bronchial tree; (2) the clinical improvement associated with treatment is characterized by a significant worsening of the histological findings; and (3) the NKA immunopositivity seems to increase significantly rather than decrease, as one would have expected, after pharmacological treatment. Further studies are necessary both to implement the number of samples and to use other markers of inflammation to characterize the potential role of cytokines in the diagnosis and therapeutic approach of severe equine asthma.

## 1. Introduction

Disorders of the respiratory system, particularly the lower airways, are the most frequently diagnosed conditions in sport horses evaluated for poor performance [1]. The terminology used to describe equine chronic noninfectious small airway disease has further evolved in the last few years with the term “equine asthma” (EA), as new features (functional, anatomical and pathobiological) of this condition are emerging [2,3]. EA is now being recommended to describe horses with chronic respiratory signs ranging in severity from mild to severe: they were previously referred as inflammatory airway disease (IAD) or recurrent airway obstruction (RAO), respectively [3]. RAO is then considered as one of the main features of horses with a severe form of equine asthma (also defined as severe equine asthma syndrome) [2,3,4,5,6], which resembles human asthma in many aspects [3,4].

Horses affected by severe equine asthma syndrome show labored breathing at rest following exposure to specific airborne agents, and reversible and reproducible airway obstruction related to the level of environmental exposures [4]. The exposure to hay and dust leading to heaves is rather a consequence of the human influence on the horses’ natural environment. Molds and fungi are indeed common antigens in stables, suggesting that EA is a disease of “domestication”. However, horses can develop a similar condition while at pasture, with grass pollen then being the likely triggering factor [7,8,9].

Clinical exacerbation occurs following exposure of susceptible horses to specific airborne agents, and results in a disease phenotype of varying severity, ranging from exercise intolerance to coughing and severe expiratory dyspnea. As the name implies, the disease is largely reversible, whereby avoidance of the inciting airborne agents results in significant disease remission over time. The cardinal clinical features of severe EA can be due to the underlying airway inflammatory response that underpins the particular functional and pathological features of the disease [10].

Since bronchoalveolar lavage (BAL) by use of fiber-optic endoscopy was first described in horses, cytological and microbiological evaluation of tracheal washes and BAL fluid have become the cornerstones in the diagnosis of respiratory disease alongside clinical and functional examinations. In exacerbation of severe equine asthma, the horses show dyspnea at rest, as revealed by a maximum intrapleural pressure >15 cm H_2_O caused by bronchoconstriction, mucosal swelling and mucus accumulation [11], and inflammation of the small airways, in which neutrophils exceed 25% in bronchoalveolar lavage fluid (BALF) cytology [2]. The definitive diagnosis of severe equine asthma is usually based on BALF cytology with the presence of lower airway inflammation, characterized by total nucleated cell count with mild increased numbers of neutrophils [12,13,14] and lymphocytes, and by increased mast cell or eosinophilic counts [15,16]. Therefore, two different cytological phenotypes are recognized in severe equine asthma: a classical neutrophilic phenotype and a paucigranulocytic phenotype. They do not correspond to the histopathological finding observable in the peripheral airway [2] and suggesting a complex role for pulmonary neutrophils in equine asthma pathophysiology. Currently, little is known about the degree and type of BALF inflammation, during the different phases of the disease (asymptomatic, exacerbation and remission phase). Moreover, pathological features of the bronchial mucosa and their value in the diagnosis of severe equine asthma syndrome are poorly characterized. To date, histological findings collected by endoscopic biopsies do not allow differentiating between controls and subjects with severe EA in remission [17,18].

The immunologic background of severe equine asthma remains not fully clarified despite many studies on the pathogenesis [19,20]. However, a positive correlation exists between the intensity of airway hyperreactivity and the quantity of chemical mediators released locally in the lung [21]. Generally, airway inflammation involves activation of a pathogen-specific inflammatory cell, the modulation of gene transcription factors, and release of inflammatory mediators [16]. Commonly accepted airway inflammatory mediators involved in airway disease include histamine, bradykinin, prostaglandin, leukotrienes, platelet-activating factor, and endothelin-1 [20,21,22]. A tachykinin mediator with a physiological and pathological role in respiratory function is a neuropeptide called neurokinin A (NKA), which is involved in the processes of bronchoconstriction and neurogenic inflammation in asthmatic patients, with potential therapeutic implications using selective NKA receptor antagonists [20]. To the best of our knowledge, only two studies investigated the role of neurokinin-A (NKA) in the horse respiratory tract [20,23], and only one by immunohistochemical methods [20].

Neutrophilic bronchiolitis is one of the main lesions of asthma-affected horse’s response to aeroallergens. Within the airways, neutrophils likely contribute to bronchoconstriction, mucus hypersecretion, and pulmonary remodeling by release of pro-inflammatory mediators, including the cytokines, among which interleukins 8 (IL-8) [16]. Determining which cytokines are implicated in the pathogenesis of EA may help in the diagnosis and treatment of this disease.

The aim of the study performed in EA-affected horses in asymptomatic, exacerbation and remission phase, is (1) to verify the diagnostic value of histology and the information deriving from it, using the histological scoring system for endoscopic biopsies, in a possible correlation with the clinical features of the different phases; (2) to analyze the immunohistochemical response for NKA and IL-8 from biopsy samples of lung tissue in subjects undergoing severe EA, in the different phases of experimentally induced disease; and (3) to evaluate the influence of sampling position along the respiratory tract on results.

## 2. Materials and Methods

### 2.1. Horses

Eleven horses were enrolled in the study, six EA-affected horses in asymptomatic phase, and five young just slaughtered horses as control, healthy at the pre-mortem visit and without evidence of pulmonary disease on post-mortem examination.

The EA-affected group consisted of five Italian Saddle horses and one Appaloosa, four mares and two geldings, 15.7 ± 1.9-year-old, with body weight ranging from 400 to 520 kg. The horses lived outdoor and had a history of signs of acute recurrent onset of asthma when housed in a stable with shavings for bedding and hay for feed. None of the horses received medications for at least three months prior to the assessment. Throughout the experimental time, a daily clinical examination was performed.

The control group comprised lungs removed immediately after slaughter from three Italian saddle horses and two Standardbreds, four mares and one male, 1.4 ± 1.5-year-old, with body weight ranging from 420 to 540 kg.

### 2.2. Experimental Design

EA-affected horses in asymptomatic phase (T0), were sedated with acepromazine maleate at 0.02 mg/kg bw iv (Prequillan, Fatro, Ozzano dell’Emilia BO, Italy) and detomidine hydrochloride at 0.01 mg/kg bw iv (Domosedan, Vetoquinol Italia, Bertinoro FC -Italy) and underwent a bronchoscopy (Pentax EG290P diameter 9.8 mm; length 120 cm).

For each endoscopic procedure performed in EA-affected horses, the instrument was introduced in the right and left principal bronchus. The airways were anesthetized by spraying of lidocaine, and four (two left and two right) epithelial biopsy specimens were obtained from four sites by use of endoscopic forceps. Biopsy sites were chosen between: 1) Right cranial bronchus and principal bronchus; 2) second right lateral segment and principal bronchus; 3) left cranial bronchus and principal bronchus; and 4) second left lateral segment and principal bronchus [24] (Figure 1).

Randomization of biopsy sequence was applied, and biopsy was performed by grabbing the lateral part of a bronchial branch with a 2.3 mm biopsy forceps, by avoiding the vessel from sliding flowing on the floor of the bronchus.

After the first bronchoscopy, the horses were brought from pasture, bedded in boxes with closed windows, on mold straw and fed hay of poor quality, to increase the dustiness of the environment and induce lower airway obstruction. Straw was not changed; it was rather raised twice a day to increase the dustiness of the environment. On day 2 (T1, early exacerbation phase) and 7 (T2, late exacerbation phase) of the challenge, bronchoscopy with bronchial biopsies was repeated with the same procedures described above. On day 8 the horses were put back to pasture after administration of dexamethasone (0.1 mg/kg) [25]. Four days later the biopsies were repeated (T3, remission phase) (Figure 2).

For the control horses, the lung was removed as soon as the horses were slaughtered and placed on a flat surface. The presence of macroscopic lung changes was evaluated; the carina was opened to the bronchial branches, to acquire biopsy samples by using a biopsy punch in the same site chosen for EA-affected horses.

### 2.3. Bronchial Biopsy Histology

Immediately after biopsy collection, the samples were fixed in 10% neutral buffered formalin. Within 24–48 h, all samples were processed as routine for histology, embedded in paraffin, cut into 5 µm-thick sections and stained with hematoxylin and eosin (HE).

The observation of histological samples provided for an evaluation according to the standardized 14-point semiquantitative grading scoring system proposed by Bullone et al., 2016 for lung pulmonary biopsies based on histological features considered important in heaves [17]. The variables evaluated and scored were: (a) epithelial hyperplasia; (b) presence of epithelial inflammatory infiltrate; (c) goblet cells hyperplasia; (d) epithelial desquamation; (e) thickening of the basal membrane; (f) submucosal inflammation; (g) the presence of mucous glands within the lamina propria; (h) the presence of mucous gland among smooth muscle bundles; (i) airways smooth muscle fibrosis; and (j) presence of the end of the smooth muscle. Each variable was measured as 0 = absent and 1 = present, and the total (0–10) was calculated for each biopsy sample.

The biopsy specimens were scored by two veterinary pathologists at high and small magnification under a light microscope (NiKonEclipse 80, Nikon Corporation, Konan, Minato-ku, Tokyo, Japan). The microphotograph images were captured using a Nikon Digital Sight SD-MS camera (Nikon Corporation, Konan, Minato-ku, Tokyo, Japan) connected to the optical microscope.

### 2.4. Immunohistochemical Analysis for Neurokinin A and IL-8

Formalin fixed paraffin embedded samples were cut into 3–5 μm sections, mounted onto poly-L-lysine coated slides, dewaxed, rehydrated, and rinsed with tap water at room temperature. Immunolabelling was performed with a streptavidin–biotin–peroxidase technique (BIO SPA, Milan, Italy). The antibodies used and their dilutions were the following:NKA (1:400, polyclonal, Biorbyt, St. Louis, MO, USA) and IL-8 (1:100 dilution, monoclonal, Cloud Clone Corporation, Katy, TX, USA). The incubation was carried out with 3% hydrogen peroxide in methanol for 30 min (to block endogenous peroxidase activity) and microwave treatment (750 W) for antigen retrieval in citrate buffer solution at pH 6.0 (one burst of 5 min and five bursts of 2 min and 30 sec, replacing the evaporated buffer after each heating session). After that, the sections were incubated overnight at 4 °C in a humid chamber with the primary antibody diluted, according to the appropriate dilutions, in PBS (0.01 M, pH 7.4). The sections, subsequently washed in PBS, were incubated first with the secondary antibody (anti-rabbit IgG conjugated with biotin) for 30 min at room temperature, and then with the streptavidin-peroxidase complex for 25 min at room temperature. After a 12-min passage in the chromogenic DAB solution (diaminobenzidine 0.02%, and H_2_O_2_ 0.001% in PBS), the sections were immediately rinsed in PBS, then in running water, stained with a contrast coloring (hematoxylin), dehydrated and assembled with DPX (Fluka, Riedel-de Häen, Germany).

NKA immunohistochemistry results were graded as follows: intensity of positivity (0, negative reaction; (1) weak intensity; (2) moderate intensity; (3) intense positivity; (4) very strong positivity); signal distribution (1, diffuse cellular, >50% of cells immunopositive; 2, outbreaks of positivity, <50% of the cells with a positive reaction), cell localization (1, cytoplasmic; 2: nuclear; 3: both cytoplasmic and nuclear).

IL-8 immunohistochemistry results were graded as follows: intensity of positivity (0, negative reaction; 1, weak intensity; 2, moderate intensity; 3, intense positivity; 4, very strong positivity); signal distribution (1, diffuse cellular, >50% of cells immunopositive; 2, outbreaks of positivity, <50% of the cells with a positive reaction).

Appropriate positive controls were used in order to evaluate the specificity of the reactions and ascertain the proper cross-reactivity in the horse tissue. Brain and normal horse lung were used as positive controls for NKA, horse lymph node as IL-8 control. As a negative control for the immunohistochemical procedure, 10% normal mouse serum was used in replicate sections instead of the primary antibody.

All the markers were evaluated through blinded observations by semiquantitative analysis of five representative high-power fields at the optical microscope (NiKonEclipse 80, Nikon Corporation, Japan). The microphotograph images were captured using a Nikon Digital Sight SD-MS camera (Nikon Corporation, Konan, Minato-ku, Tokyo, Japan) connected to the optical microscope.

### 2.5. Statistical Analysis

Statistical analysis was performed with a commercially available program (GraphPad Prism 5, GraphPad Software, San Diego, CA, USA). Assessment of data for normality was calculated by applying the D’Agostino-Pearson test. Data were expressed as median (minimum-maximum).

Differences in histological score, NKA and IL-8 immunohistochemistry between four bronchial sampling sites were analyzed by a Kruskal–Wallis one-way analysis of variance, both in EA-affected horses (at each experimental time), and in control horses. 

For each parameter examined in each experimental time, the median value of variables of four biopsy samples was used for the subsequent statistical analysis.

A Mann–Whitney test (two tail *p* value) was applied to perform a comparison between control horses and EA-affected horses, at each experimental time, for the following variables: (1) Histological score; (2) NKA immunohistochemistry score; and (3) IL-8 immunohistochemistry score. Similarly, a Friedman test with Dunn’s multiple comparison test, as post hoc test, was applied to perform a comparison of EA-affected horses at different experimental times, for the following variables: (1) Histological score; (2) NKA immunohistochemistry score; and (3) IL-8 immunohistochemistry score.

The significance was set for *p* < 0.05.

## 3. Results

All the horses, initially asymptomatic, showed respiratory signs related to equine asthma exacerbation, after the transition from the pasture to the stable. Indeed, after 2 days from the beginning of the test (T1), clinical examination showed cough and dyspnea in all subjects. However, dyspnea showed a quick remission after environmental change and drug treatment at T3.

Histological score, and NKA and IL-8 immunohistochemistry scores of bronchial biopsies revealed no significant differences between the different biopsy sites nor in EA-affected horses in every experimental time, and in slaughtered horses (Table 1).

With the exception of the first sampling performed, endoscopic biopsies in EA-affected horses were found to be evaluable in at least one of the four samples each horse in each phase, and therefore samples of all phases in all cases were included in the statistical evaluation, both for histology and for immunohistochemistry.

The histopathology evaluation of the bronchial samples collected in the slaughtered horses ranged from a score of 2–3, allowing to consider those subjects as not affected by any respiratory problem, and to include them all as control-cases (see Supplementary Material Appendix A). The histological score of EA-affected horses is detailed in Supplementary Material, Appendix A. Relevant pictures of histological features are reported in Figure 3.

The comparison of histological scores of control vs EA-affected horses at four experimental points (T0; T1; T2; T3) demonstrated a significant difference at T0 (*p* = 0.001), in the asymptomatic phase; at T1 (*p* = 0.007), in the early exacerbation phase; at T2 (*p* = 0.007), in the late exacerbation phase; and at T3 (*p* = 0.007), in the remission phase (Table 2).

Median histological scores of bronchial biopsies of EA-affected horses during the different experimental phases (T0, T1, T2, T3), evidenced a significant increase moving from the asymptomatic phase (T0) to the remission phase (T3) with significant difference between T0 and T3 (*p* = 0.01) (Table 3).

The results for each immunohistochemical marker are reported in Table 4 and Table 5.

Relevant pictures of the IHC on control and EA-affected horses for NKA and IL-8 are reported in Figure 4 and Figure 5, respectively.

In control horses, NKA immunolabelling was weak, widespread cytoplasmic in the epithelial bronchial cells. In EA-affected horses, NKA immunolabelling appeared moderate to strong staining at the cytoplasm and occasionally in the nucleus (was found in a minority of epithelial cells also in the nucleus) (Figure 4c,d). In many cases, the immunopositivity was confined to the apical portion of epithelial cells of the airway mucosa (Figure 4e).

The expression of IL-8 was cytoplasmatic. IL-8 positivity was seen in epithelial cells of control horses with a discontinuous, focal, weak staining. IL-8 immunolabelling in equine asthma instead appeared multifocal to diffuse, cytoplasmic, and moderate to intense (Figure 5b,c), in some cases confined to isolated respiratory epithelial cells (Figure 5e,f).

NKA immunopositivity scores of bronchial biopsies showed a significant increase of intensity of positivity, moving from control horses to EA-affected horses at T2 (*p* = 0.04), and at T3 (*p* = 0.04), while no significant differences were showed for signal distribution and cell localization between groups (Table 2). No differences for NKA immunohistochemistry results (intensity of positivity, signal distribution, cell localization), were recorded in EA-affected horses, among the experimental times (Table 3).

At last, IL-8 immunopositivity score, subdivided on intensity of positivity and signal distribution, recorded no significant differences between control horses and EA-affected horses (Table 2), nor in EA-affected horses in each experimental time (Table 3).

## 4. Discussion

The diagnosis of severe equine asthma (EA) is mainly based on history and clinical signs [2,3,6,25]. Collateral diagnostic investigations with additional tests, such as an airway endoscopy or lung function evaluation, can confirm and further characterize the diagnostic suspect of severe equine asthma. Among these additional tests, the cytological examination of bronchoalveolar lavage fluid (BALF) is still considered as a ‘gold standard’ in the diagnosis of severe lower airway inflammation in the horse [2,6,16,18].

However, if on the one hand the BALF cytology is able to disclose neutrophilic inflammation during the exacerbation of the disease [14], on the other hand, the inflammatory pattern completely normalizes during periods of remission, induced by antigen avoidance strategies [17,26]. Clinical signs are also largely reversible after treatment with inhaled or systemic corticosteroids, in absence of harmful environment [2,17]. Conversely, remodeling of the peripheral airways in the severe EA persists during the remission phases and it is related to residual airflow obstruction [26].

Airway remodeling includes increased airway smooth muscle mass, goblet cell hyperplasia/metaplasia, collagen and elastic fiber deposition within the lamina propria, peribronchiolar fibrosis, airway obstruction by mucus and inflammatory cells, airway adventitial inflammation [17,27]. Therefore, it is not yet clear which role the above-mentioned remodeling process plays and how much it may affect airways inflammation and especially in maintaining disease during asymptomatic periods. Moreover, due to the absence of a significant association between lower airway inflammation detected with BALF and the degree of pulmonary dysfunction in equine asthma, the study of the structural small and large airway alterations has recently gained interest [2]. Identifying specific pathologic variables by endoscopic biopsy specimens could possibly facilitate the reaching of a diagnosis during the remission and monitoring response to treatment [17].

The histopathological characteristics of the bronchial mucosa and their evaluation in the diagnosis of severe EA are currently poorly characterized and very few information concerning the histological evaluation of endobronchial biopsies of the horse is present in the literature [17,18,28]. The main reasons for this lack consist of the fact that the transthoracic biopsy samples are a risky and expensive procedure, and therefore not carried out in clinical practice. The only studies existing in the literature for the histological evaluation of peripheral airway remodeling concern histological studies on lungs of horses during autopsy or slaughter [27,29]. On the contrary, the only method that can be used in life is an endobronchial biopsy sampling from the central airways, which is quite easy and repeatable. Nevertheless, the biopsies obtained from the large respiratory tract do not specifically collect material from the areas affected by the pathology; the information is still very scarce in the course of EA [2]. If the effects were similar, you could then be able to make a diagnosis during the clinical remission phase of the severe EA. Furthermore, the studies carried out on BALF and on endobronchial biopsy specimens have focused on biomolecular type investigations gene expression [5,30,31,32,33,34,35,36,37,38,39,40]. Contrariwise, only very few papers deal with biopsies samples on paraffin sections for histological and immunohistochemical evaluations [5,17,18,20,37].

In our procedure, the difficulties in bioptic procedures encountered in the first sampling (EA-affected 3), suggested by the classic methodologies, which made the sampled material too superficial and unsuitable for evaluation, were largely solved using a manual technique that provided for access to the wall of the bronchus, by withdrawing the forceps towards the endoscope and at the same time by exerting a twist in such a way as to acquire a larger sample.

Bullone et al., 2016 first developed a 14-point histological scoring system that assesses histological variables of inflammation and remodeling of the large airways using endoscopic biopsy specimens [17]. This evaluation system, however, although considered a reliable tool for the assessment of airway obstruction caused by inflammation and remodeling in horses with severe forms of EA, still requires a standardized evaluation and is currently rarely used [3,18]. Their application of this scoring system in 20 horses (controls, exacerbation and remission of heaves) disclose a correlation between histology and the degree of airway obstruction clinically measured, but does not seem to discriminate horses with heaves in remission from control [17].

Using a classical histologic assessment for evaluation of endoscopic biopsies, Niedzwieds et al., 2018 also found no significant difference for any of the histological variables between control and study group with asthma, and concludes hoping for the use of a standardized scoring system, such that developed by Bullone et al., 2016, to differentiate horses with an asthma exacerbation from healthy horses [17,18].

In our study, we chose to use the same scoring system to Bullone et al., 2016 for histologic evaluation of biopsy samples [17]. Conversely to previously results, we found significant differences in histological scoring between subjects in remission and healthy control horses. These results demonstrate how, similarly to what happens for the remodeling of the peripheral airways, the progression and maintenance of the disease are preserved histologically even in the remission phase, and an altered immunoregulatory function of the airway epithelium in horses with severe equine asthma syndrome.

The most common mediators of airway inflammation involved in horse respiratory diseases include histamine, bradykinin, prostaglandins, leukotrienes, platelet factor and endothelin-1 [17,40,41]. Furthermore, another mediator with a physiological and pathological role in respiratory function is a neuropeptide called neurokinin A (NKA), a member of the tachykinin family [42,43,44]. Neurokinin A is involved in nonadrenergic/noncolinergic neurotransmitter mediation of excitatory type in the respiratory tract of rat and guinea pig in normal conditions [45,46]. The NKA effect manifests through the activation of NK-2 receptors; they contract the smooth muscle of the airways leading to narrowing of the lumen. An abnormal expression of NK-2 receptors has been associated in human beings with forms of asthma [47,48].

On the one hand, the role of NKA in bronchoconstriction and neurogenic inflammation in patients with asthma generated scientific attention and specific investigations on selective NKA receptor antagonists for therapeutic uses [49]. On the other hand, Neurokinin B has been shown to be a much less potent contractile agent in vitro than NKA in both control and EA horses, probably due to fewer receptors or less affinity [20]. However, little information exists regarding the involvement of NKA receptors in the horse, moreover limited to the intestine [50] and to the lung of horses affected by EA [20]. Recently, the receptors of the capsaicin sensitive-sensory nerves stimulated by bacterial of lipopolysaccharides, capable of inducing neurogenic inflammation [23] as occur in human airways [48] have been investigated in equine bronchial tissue.

The research conducted by Venugopal et al., 2009 on healthy and EA-affected horses through a comparative method demonstrates how significantly the expression of NK-2 has increased in the bronchial epithelium and in the bronchial smooth muscle and lung vessels of EA-affected horses [20]. The results of this study demonstrate how much NK-2 receptors are upregulated in EA, and suggest that NK-2 receptor antagonists may have some therapeutic effect in controlling the progression of airway hyperreactivity. Based on the theory that both NKA and NKB can cause concentration-dependent contractions in horse bronchial rings, and that this response is more evident in horses affected by severe equine asthma, the Authors hypothesized that the increase in the expression of NK-2 receptors may be a major cause of airway hyperreactivity in EA-affected horses.

The ex vivo study by Calzetta et al., 2018 confirms and extends the previous results by Venugopal et al., 2009 and suggests that the environmental exposure to bacterial of lipopolysaccharides may represent a crucial factor in modulating the bronchial responsiveness of severe equine asthma [20,23]. A chronic exposure to high concentration of inhaled endotoxin in fact has a deleterious effect in equine distal airways: it triggers dysfunctional airway smooth muscle (ASM) contractility due to the stimulation of capsaicin-sensitive sensory nerves, increased the release of NKA and the activation of NK2 receptors. Antagonizing NK2 receptors may have a beneficial impact on normalizing the contractility and on controlling clinical signs in severe equine asthma.

Our results reveal a significant variability in the immunohistochemical expression of NKA between control horses and horses with severe equine asthma in remission. However, contrarily to what was expected, NKA immunoreactivity appears to increase significantly after corticosteroid treatment. This demonstrates to what extent the clinical improvement associated with therapy does not correspond to a decrease in this marker of inflammation, but rather show a contrary trend. Therefore, this result could suggest that, unlike cortisone drugs, there are potential bases for an efficacy of selective and non-selective NK2 receptor treatments.

T lymphocytes play a fundamental role in modulating the immune response during the pathogenesis of severe equine respiratory syndrome. Data of the literature suggests that lung T helper may be implicated through the secretion of Th-1 or Th-2 type cytokines [19,37,38,51,52,53,54]. Horses affected by severe equine asthma syndrome produce both of these cytokines, depending on the stage of the disease and the time when sampling is performed [55]. In addition, the expression of cytokines in the airway lymphocytes also appears to be influenced by the length of time in which the horses have manifested the disease clinically [55].

The bronchiolar intraluminal accumulation of neutrophils usually accompanies the lymphocyte peribronchiolar infiltrate in severe EA anatomopathological findings [56], and takes place a few hours after environmental changes. The late phase of type I hypersensitivity plays a central role in a Th mixed response, mainly Th2 at the beginning and Th1 in the late phase of the process [35]. Both ways contribute to triggering inflammation and neutrophil recruitment in the airways by means of several cytokines, such as IL-8 [37,52,57]. In addition, a type III hypersensitivity reaction is able to partially explain the neutrophilic inflammation in the airways on EA-affected horses. However, the factors that initiate this process are still not fully explained [55].

Studies on horses with severe equine respiratory syndrome have revealed an increase in the gene expression of the pro-inflammatory cytokine IL-8 in bronchoalveolar cells [19,38,40,52,58,59], as well as an increase in IL-8 protein concentration in BALF [19,37,38,40,60,61,62,63,64,65].

It is well known that the main producers of IL-8 are the airway epithelial cells [27]. Ainsworth et al., 2006 demonstrated a 3- to 10-fold increase of IL-8 m-RNA in epithelial airway cells in severe equine asthma, and afterwards Berndt et al., 2007 confirmed how IL-8 mRNA expression in bronchial epithelial cells increases after exposure to stable dust, both in EA-affected subjects and in control animals [27,61]. These results were also confirmed by another study [38], in which an increase in the level of IL-8 mRNA is observed both in BAL cells and in endobronchial biopsies during the crisis phases of the disease.

On the contrary, in the study by Pietra et al., 2011, although a significant overexpression of IL-8 and TNFα mRNA on BALF was observed in the group of treated horses, with a peak around the ninth day, a comparison between exposed and unexposed horses does not reveal significant differences [64]. According to the authors, this evidence could depend on the loss of uniformity of the sample, since the evolution of the pathology represents a continuum from healthy to pathological conditions.

Padoan et al., 2013, in a comparative research between clinical and endoscopic findings, cytological and cultural tests of BALF, histology of bronchial tissue and analysis of gene expression of inflammatory mediators in BALF and biopsies, found that six out of ten immunologically related genes (including IL-8) show a significant difference in expression between horses affected by severe equine asthma syndrome and the control group [40]. Specifically, however, IL-8 mRNA levels measured in biopsies are a hundred times lower than in BALF, and although an increase in its gene expression has been found in bronchial biopsies, this difference is not statistically relevant. Furthermore, the biopsies included in the study showed no significant differences in gene expression levels between EA-affected horses and controls. The lack of statistical significance between mRNA levels of inflammatory mediators in the respiratory epithelium could be caused by the small size of the samples or by the site where the biopsy is performed.

By using immunohistochemistry, Ainsworth et al., 2006 demonstrated IL-8 localization in the cytoplasm of epithelial cells in airway biopsies as well as in more recent times Tessier et al., 2017 observed a marked increase in IL-8 gene product determined by immunohistochemistry in bronchial cells in asthmatic, but not in non-asthmatic animals. In our study, we found significant differences between healthy and EA-affected in epithelial immunohistochemical expression, but none in IL-8 immunohistochemistry expression in the different stages of EA [5,37]. Unlike former suggestions in literature data on biomolecular BALF evaluation, however, this result appears to be aligned with the histological evaluations, which do not reveal any relevant difference in the inflammatory evaluation parameters in all phases.

Beyond the statistical significance of the results, the immunohistochemical observations reveal however interesting suggestions: the reactivity to NKA shows a trend to increase in intensity and to show same dysregulation characteristics (with the appearance of abnormal nuclear locations and variations in the mucosal distribution) in horses with severe equine syndrome compared to those of control. For IL-8, in horses with EA there is an intense positivity—often focal and localized to in single cells—which instead is not present in all the control subjects, where positivity is always widespread and of far weaker intensity. As far as these observations have diagnostic and/or immunopathogenic relevance for this disease in its various stages, it is not currently possible to determine it with certainty. Moreover, studies in this regard require a number of samples significantly higher and accompanied by biomolecular analyzes.

## 5. Conclusions

The results of the present study demonstrate homogeneity in the bronchial endoscopic samples regardless of the sampling site, demonstrating a lack of influence of bronchial sampling position on histological and immunohistochemical evaluation. The application of the histological score of Bullone et al., 2016 to endoscopic biopsies [17], however, fails to first differentiate between the exacerbation and the remission phase, which, contrarily to what we found, should have a return to histological conditions significantly similar to those of normality.

By immunohistochemical analysis, the comparison between clinically healthy horses and EA-affected ones suggested new insights on the cytokine expression in equine health and disease status. On the one hand, investigations on endoscopic biopsy samples allowed detecting relevant differences in immunoreactivity between control horses and horses with severe equine asthma syndrome, demonstrating a dysregulation of their expression in the latter. On the other hand, the clinical improvement at different times of the disease seems to have no immunohistochemical bases for IL-8 and NKA expression. The explanations behind these results are still being studied and require more data and subjects examined.

## Figures and Tables

**Figure 1 animals-11-01376-f001:**
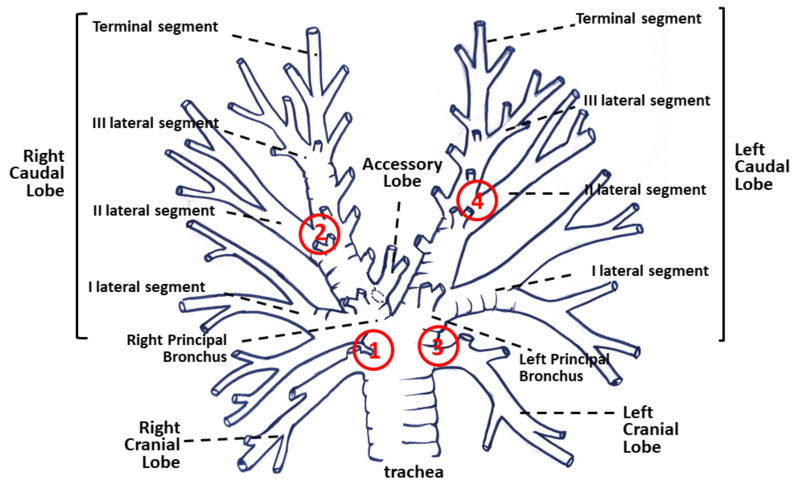
Position of bronchial biopsies: (1) Between right cranial bronchus and principal bronchus; (2) between second (II) right lateral segment and principal bronchus; (3) between left cranial bronchus and principal bronchus; and 4) between second (II) left lateral segment and principal bronchus.

**Figure 2 animals-11-01376-f002:**
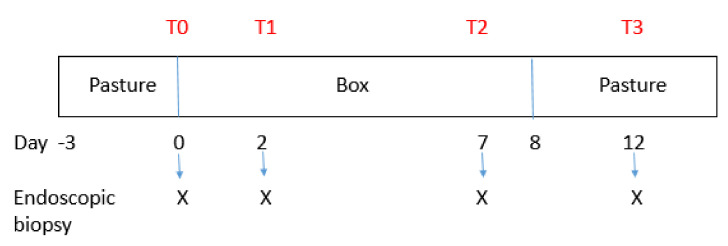
Timeline flow chart of trial.

**Figure 3 animals-11-01376-f003:**
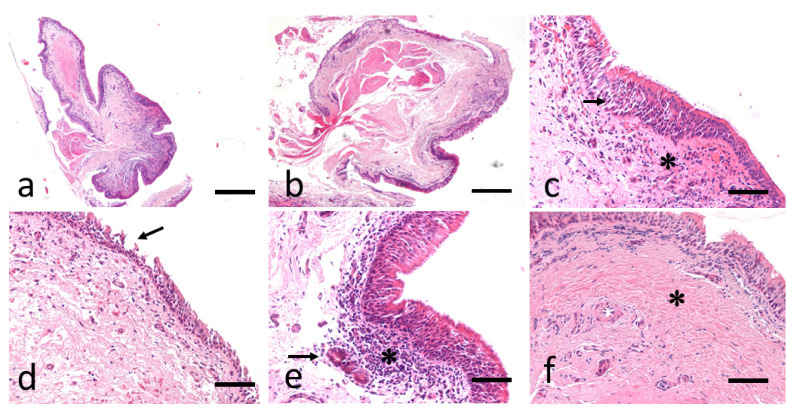
Examples of histological samples of bronchial biopsy in the EA horses. (**a**,**b**). Low-magnification images of biopsies, obtained with tissue twisting technique with biopsy forceps before tearing. All tissue layers are observed (surface epithelium, extracellular matrix and airway smooth muscle). HE, bar 800 µm. (**c**) Marked mucosal hyperplasia (arrow), and the presence of a moderate inflammatory submucosal infiltrate of lymphocyte (asterisk). HE, bar 200 µm. (**d**) Epithelial desquamation involving over 70% of the mucous layer (arrow). HE, bar 200 µm. (**e**) Severe diffuse lymphocytic infiltrate (asterisk) and presence of mucosal gland (arrow). HE, bar 200 µm. (**f**) Severe submucosal fibrosis (asterisk). HE, bar 200 µm.

**Figure 4 animals-11-01376-f004:**
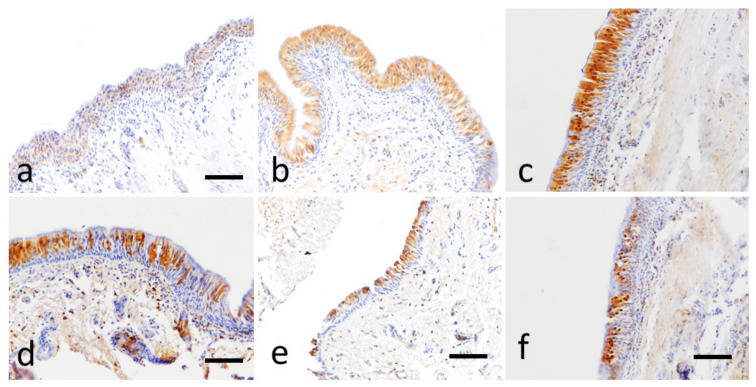
NKA immunohistochemical stain (brown color), bar 200 µm. (**a**) Bronchial mucosa of a control horse. Weak and diffuse cytoplasmic positivity of the respiratory epithelium. (**b**) EA-case 3, distal right bronchus, T2. Moderate and diffuse cytoplasmic positivity to the epithelial cells of the respiratory mucosa, associated with an intense nuclear positivity. (**c**) EA-case 1 horse, proximal right bronchus, T3. Intense, diffuse, cytoplasmic and nuclear positivity, mainly localized in the more superficial layers of the hyperplastic respiratory epithelium. (**d**) EA-case 3 horse, distal right bronchus, T3. Intense, diffuse cytoplasmic and nuclear localized mainly in the superficial layers of the epithelium. (**e**) EA-affected 6, proximal left bronchus, T1. Positivity confined to the luminal layer of the epithelium. (**f**) EA-case 2, distal left bronchus, T3. Predominantly nuclear positivity.

**Figure 5 animals-11-01376-f005:**
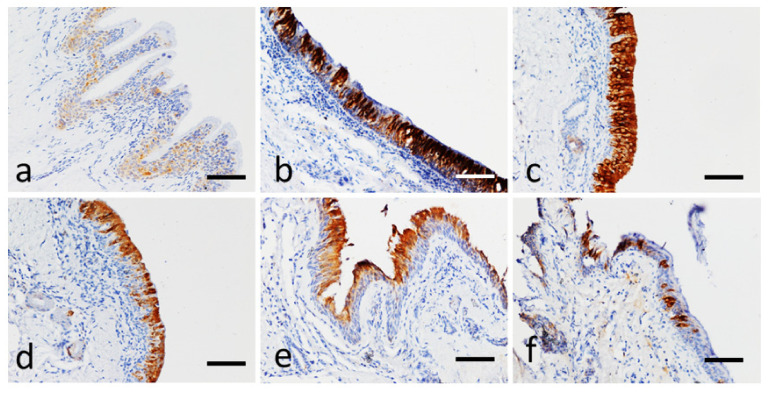
IL-8 immunohistochemical stain (brown color), bar 200 µm. (**a**) Bronchial mucosa of a control horse. Focal and weak cytoplasmic positivity to the respiratory epithelium. (**b**) EA-case 2, distal right bronchus, T3. Strong and diffuse positivity to the epithelial cells of the mucosa. (**c**) EA-case 2, proximal right bronchus, T3. Note the intense immunopositivity in all layers of the hyperplastic mucosa. (**d**) EA-case 2, distal right bronchus, T2. Note the distribution of positivity in the most superficial layers of the mucosa. (**e**) EA-affected horse 6, proximal left bronchus, T3. The positivity appears to be confined to the most superficial layer of the epithelium also in this case. (**f**) EA-case 5, proximal left bronchus, T1. Focal positivity confined to isolated respiratory epithelial cells.

**Table 1 animals-11-01376-t001:** Comparison of histological, NKA immunoreactivity score and IL-8 immunoreactivity score acquired from bronchial biopsies, obtained in four standardized position, from control horses and EA-affected horses at T0, T1, T2, T3. Median value (min-max) and *p* value—Kruskal–Wallis test.

	Control	EA-Affected	EA-Affected	EA-Affected	EA-Affected
		T0	T1	T2	T3
**Histological score (0–10)**	3	5.5	5	6.5	7.5
median (min-max)	(2–4)	(4–6)	(4–8)	(5–10)	(5–9)
*p*	*p* = 0.92	*p* = 0.48	*p* = 0.16	*p* = 0.63	*p* = 0.81
**NKA immunoreactivity score**					
Intensity of positivity (0–4)	2	3	3	3	4
median (min-max)	(2–3)	(1–4)	(2–4)	(2–4)	(2–4)
*p*	*p* = 0.5	*p* = 0.88	*p* = 0.49	*p* = 0.83	*p* = 0.06
Signal distribution (1–2)	1	1	2	1	1
median (min-max)	(1–1)	(1–2)	(1–2)	(1–2)	(1–2)
*p*		*p* = 0.27	*p* = 0.99	*p* = 0.54	*p* = 0.91
Cell localization (1–3)	1	1.5	1	1.5	2
median (min-max)	(1–1)	(1–3)	(1–3)	(1–3)	(1–3)
*p*		*p* = 0.58	*p* = 0.96	*p* = 0.41	*p* = 0.85
**IL-8 immunoreactivity score**					
Intensity of positivity (0–4)	2	3	3	3	3
median (min-max)	(1–3)	(3–4)	(2–4)	(2-4)	(2-4)
*p*	*p* = 0.07	*p* = 0.65	*p* = 0.87	*p* = 0.35	*p* = 0.34
Signal distribution (1–2)	2	1	2	1	1
median (min-max)	(1–2)	(1–2)	(1–2)	(1–2)	(1–2)
*p*	*p* = 0.5	*p* = 0.56	*p* = 0.63	*p* = 0.48	*p* = 0.41

**Table 2 animals-11-01376-t002:** Comparison of median value (min-max) of histological, NKA and IL-8 immunoreactivity scores acquired from bronchial biopsies, between control horses and EA-affected horses at T0, T1, T2, T3. Mann–Whitney test. a,b: different letters indicate significant differences (*p* < 0.05); A, B: different letters indicate significant differences (*p* < 0.01).

	Control	EA-Affected	EA-Affected	EA-Affected	EA-Affected
		T0	T1	T2	T3
**Histological score (0–10)**	3	5.5	5	6.5	7.5
median (min-max)	(2–4) A	(4–6) B	(4–8) B	(5–10) B	(5–9) B
**NKA immunoreactivity score**					
Intensity of positivity (0–4)	2	3	3	3	4
median (min-max)	(2–3) a	(1–4)	(2–4)	(2–4) b	(2–4) b
Signal distribution (1–2)	1	1	2	1	1
median (min-max)	(1–1)	(1–2)	(1–2)	(1–2)	(1–2)
Cell localization (1–3)	1	1.5	1	1.5	2
median (min-max)	(1–1)	(1–3)	(1–3)	(1–3)	(1–3)
**IL-8 immunoreactivity score**					
Intensity of positivity (0–4)	2	3	3	3	3
median (min-max)	(1–3)	(3–4)	(2–4)	(2–4)	(2–4)
Signal distribution (1–2)	2	1	2	1	1
median (min-max)	(1–2)	(1–2)	(1–2)	(1–2)	(1–2)

**Table 3 animals-11-01376-t003:** Comparison of median value (min-max) of histological, NKA and IL-8 immunoreactivity scores acquired from bronchial biopsies, between EA-affected horses at T0, T1, T2, T3. Friedman test with Dunn’s multiple comparison test as post hoc test. a,b: different letters indicate significant differences (*p* < 0.05).

	EA-Affected	EA-Affected	EA-Affected	EA-Affected
	T0	T1	T2	T3
**Histological score (0–10)**	5.5	5	6.5	7.5
median (min-max)	(4–6) a	(4–8)	(5–10)	(5–9) b
**NKA immunoreactivity score**				
Intensity of positivity (0–4)	3	3	3	4
median (min-max)	(1–4)	(2–4)	(2–4)	(2–4)
Signal distribution (1–2)	1	2	1	1
median (min-max)	(1–2)	(1–2)	(1–2)	(1–2)
Cell localization (1–3)	1.5	1	1.5	2
median (min-max)	(1–3)	(1–3)	(1–3)	(1–3)
**IL-8 immunoreactivity score**				
Intensity of positivity (0–4)	3	3	3	3
median (min-max)	(3–4)	(2–4)	(2–4)	(2–4)
Signal distribution (1–2)	1	2	1	1
median (min-max)	(1–2)	(1–2)	(1–2)	(1–2)

**Table 4 animals-11-01376-t004:** Results of immunohistochemical evaluation to NKA of all endoscopic samples in the six EA-affected horses in all phases of experimental trial (T0, T1, T2, T3).

NKA	EA-Case 1				EA-Case 2				EA-Case 3				EA-Case 4				EA-Case 5				EA-Case 6			
	B Right		B Left		B Right		B Left		B Right		B Left		B Right		B Left		B Right		B Left		B Right		B Left	
	P	D	P	D	P	D	P	D	P	D	P	D	P	D	P	D	P	D	P	D	P	D	P	D
T0	n.e.	n.e.	n.e.	n.e.	2, N, diffuse	2, C, diffuse	3, N, diffuse	2, C, diffuse	n.e.	n.e.	n.e.	n.e.	2, C, diffuse	3, CN, diffuse	n.e.	2, N, diffuse	3, C, focal	n.e.	n.e.	3, CN, diffuse	1, C, focal	n.e.	1, C, diffuse	n.e.
T1	n.e.	2, C, diffuse	n.e.	2, C, diffuse	1, C, diffuse	3, N, diffuse	2, C, diffuse	3, CN, diffuse	2, CN, diffuse	3, CN, diffuse	2, CN, diffuse	3, CN, diffuse	1, C, focal	1, C, focal	1, C, focal	1, C, focal	n.e.	3, C, focal	3, C, focal	3, C, focal	1, CN, focal	2, CN, focal	3, CN, focal	2, C, focal
T2	n.e.	2, C, diffuse	n.e.	3, CN, diffuse	2, C, diffuse	2, C, diffuse	2, C, diffuse	1, C, diffuse	2, C, diffuse	3, CN, diffuse	2, C, diffuse	3, CN, diffuse	1, C, focal	2, CN, diffuse	2, C, diffuse	3, CN, diffuse	2, CN, focal	3, CN, diffuse	2, C, focal	3, N, focal	2, CN, focal	1, CN, focal	1, C, focal	1, C, focal
T3	3, N, diffuse	3, N, diffuse	3, N, diffuse	2, N, diffuse	n.e.	3, CN, diffuse	3, CN, diffuse	1, C, diffuse	3, CN, diffuse	3, CN, diffuse	3, C, diffuse	3, C, diffuse	3, CN, focal	n.e.	3, CN, diffuse	3, CN, diffuse	3, CN, diffuse	3, CN, focal	3, N, focal	2, CN, focal	1, C, focal	n.e.	n.e.	1, C, focal

B: bronchus, P: proximal, D: distal; C: cytoplasmic immunopositivity; N; nuclear immunopositivity; n.e.; not evaluable.

**Table 5 animals-11-01376-t005:** Results of immunohistochemical evaluation to IL-8 of all endoscopic samples in the six EA-affected horses in all phases of experimental trial (T0, T1, T2, T3).

IL-8	EA-Case 1				EA-Case 2				EA-Case 3				EA- Case 4				EA- Case 5				EA- Case 6			
	B Right		B Left		B Right		B Left		B Right		B Left		B Right		B Left		B Right		B Left		B Right		B Left	
	P	D	P	D	P	D	P	D	P	D	P	D	P	D	P	D	P	D	P	D	P	D	P	D
T0	3, focal	n.e.	3, focal	n.e.	3, diffuse	4, diffuse	4, diffuse	2, diffuse	n.e.	n.e.	n.e.	n.e.	3, focal	4, diffuse	n.e.	3, diffuse	4, diffuse	3, focal	n.e.	3, diffuse	n.e.	n.e.	n.e.	n.e.
T1	n.e.	2, focal	n.e.	3, focal	3, diffuse	3, focal	3, focal	4, diffuse	n.e.	3, diffuse	4, diffuse	3, focal	2, focal	3, focal	2, focal	3, focal	4, diffuse	4, diffuse	4, diffuse	4, diffuse	4, diffuse	4, focal	3, focal	4, diffuse
T2	n.e.	4, diffuse	3, diffuse	3, focal	3, diffuse	3, diffuse	3, focal	4, diffuse	2, focal	2, diffuse	2, focal	2, diffuse	2, focal	3, diffuse	3, diffuse	3, focal	3, diffuse	3, diffuse	4, focal	2, diff	2, focal	4, focal	3, diffuse	2, focal
T3	4, diffuse	4, diffuse	3, diffuse	3, focal	n.e.	n.e.	4, diffuse	4, diffuse	3, diffuse	4, diffuse	3, diffuse	4, diffuse	4, diffuse	n.e.	3, diffuse	3, diffuse	4, diffuse	4, focal	4, focal	n.e.	n.e.	3, focal	2, focal	2, focal

B: bronchus, P: proximal, D: distal; n.e.; not evaluable.

## Data Availability

All data generated or analyzed during this study are included in this published article. The raw datasets used and analyzed during the current study are available from the corresponding author on reasonable request.

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
