# Peer review of "Immunohistochemical Expression of Neurokinin-A and Interleukin-8 in the Bronchial Epithelium of Horses with Severe Equine Asthma Syndrome during Asymptomatic, Exacerbation, and Remission Phase"

_animals, 2021, doi:10.3390/ani11051376_

Round 1

Reviewer 1 Report

Dear Authors, please change all marked "RAO" to the correct term "asthma". The term "asthma" in horses with heaves replaced the old term RAO!

Reviewer 2 Report

Line 257: did you want to write median instead of mean?

Table S1 (Supplementary material): In the last column "ASM ending visible" it's not clear when you write sì

Reviewer 3 Report

The aim of manuscript “Immunohistochemical Expression of Neurokinin-A and Inter-2 leukin-8 in the Bronchial Epithelium of Horses with Recurrent Airway Obstruction During Asymptomatic, Exacerbation and Remission Phase” was to analyze, was to investigate the histological features, and the Neurokinin-A (NKA) and Interleukin-8 (IL-8) immunoreactivity on bronchoscopic biopsies in horses, obtained during different phases of the disease (asymptomatic, exacerbation and remission).

It is an interesting work, well explained, with sections presented in a balanced and coherent way. The introduction, although long, provides a good picture of the situation and properly documented. The Tables are very informative, the photomicrographs are very good (the captions can be improved, please see my comments). The Results are properly discussed. I suggest moderate revisions, nothing conceptual, it has more to do with information that I think can be added, in a constructive perspective.

Comment 1, Abstract (lines 28): I suggest “Severe equine asthma syndrome (RAO) …”

Comment 2, Introduction (line 84): I suggest “…exceed 25% in BAL fluid (BALF) cytology [2]” or “…exceed 25% in bronchoalveolar lavage fluid (BALF) cytology [2]”

Comment 3, Material and Methods (line 130): Were the 5 young horses used as controls slaughtered for this research, or were they going to be slaughtered for other reasons and were the lungs used?

Comment 4, Material and Methods: At what period of the year was the experiment performed?

Comment 5, Material and Methods (Figure 1): It is possible to improve the definition of Figure 1?

Comment 6 Material and Methods (line 142): As the authors do for the asymptomatic phase (T0), it would also be interesting to identify the other phases T1… T3, with the respective pathophysiological condition, exacerbation, remission phase, etc in this section. Or in the Results, at least the first time that each of them appears in the text, or in footnotes of the Tables.

Comment 7, Material and Methods (lines 161-164): I think it would be more explicit if in the text the authors said that the 1st biopsy was performed on day 0 (T0) after the horses arrived from the pasture to the box/stable.

Comment 8, Material and Methods (lines 176-190): How were the slides observed? How many fields? Magnification? How were photomicrographs obtained?

Comment 9, Material and Methods (line 192): I suggest start the sentence with “Formalin fixed paraffin embedded samples were cut into 3-5 μm sections, mounted onto gelatin or poly-L-lysine coated slides (?), dewaxed, rehydrated and rinsed with tap water at room temperature.”

Comment 10, Material and Methods (lines 200-213): There seems to be duplication of the text.

Comment 11, Material and Methods (line 222): I suggest writing “The expression of IL8 is cytoplasmatic”, or, in Table 5 (line 339). On the other hand, in Table 5 (line 339) the “C, Cytoplasmic” it doesn't seem to make sense because it does not appear in the description of the results.

Comment 12, Material and Methods (lines 191-227): How were the slides observed/analyzed? How many fields? Magnification? How were photomicrographs obtained? Blindly?

Comment 13, Results (Figure 3, line 306): As a histologist I understand what the authors refer to in the caption, but for other readers I think it would be important to put some notations in the photomicrographs (arrows, asterisks, etc.). The same for the legends of Figures 4 and 5.

Comment 14, Results (Table 5): Please correct title of Table 5 to IL8

Comment 15, Results (Figure 4 and 5): As I said in Comment 13, I understand what the authors refer to in the caption, but for other readers, in this particular case for readers less familiar with IHC, I think it would be important to put some notations in the photomicrographs: or arrows, asterisks, etc.), or, for example “Figure 4. NKA immunohistochemical stain (brown color)…”

“Figure 5. IL-8 immunohistochemical stain (brown color)…”

Comment 16, Conclusions (line 595): “… sampling position on histological and immunohistochemical evaluation/analys/results” or “… sampling position on histology and immunohistochemistry”.

Comment 17, Conclusions (line 606): “…seems to have no immunohistochemical bases for IL-8 and NKA expression”.
